# Schistosomiasis and Recurrent Arthritis: A Systematic Review of the Literature

**DOI:** 10.3390/pathogens11111369

**Published:** 2022-11-17

**Authors:** Coline Mortier, Maïssane Mehadji, Sophie Amrane, Anne-Laurence Demoux, Coralie L’Ollivier

**Affiliations:** 1IHU-Méditerranée Infection, Assistance Publique Hôpitaux de Marseille (AP-HM), 13005 Marseille, France; 2Assistance Publique Hôpitaux de Marseille, Service de médecine interne, Hôpital Nord, 13015 Marseille, France; 3Aix Marseille University, IRD, AP-HM, SSA, VITROME, IHU Méditerranée, 13005 Marseille, France

**Keywords:** schistosomiasis, *Schistosoma haematobium*, arthritis, parasitic rheumatism, bilharzial arthropathy

## Abstract

Background. Schistosomiasis is an endemic parasitic infection found in many tropical countries and is highly prevalent in sub-Saharan Africa. It can follow different and atypical clinical patterns. In these unusual cases, diagnosis may be difficult, as symptoms are unspecific. Arthropathy can appear in parasitic infections, but making a connection between arthritis and parasitic aetiology is difficult. This review aims to summarise all cases that have reported schistosomiasis associated with arthropathy, and the different ways authors have diagnosed this disease. Method. We present a systematic literature review of schistosomiasis associated with joint impairments, with a focus on the difficulty of differentiating between reactive arthritis and its parasitic presence in situ. Results. Joint impairments mimicking polyarthropathy are not rare in parasitic infections. Diagnosis is difficult. On the one hand, some patients have arthritis with parasite eggs found in situ, particularly in synovial biopsy. These situations are less common and antiparasitic treatment is straightforward. On the other hand, arthritis can be associated with parasitic infections in the form of reactive arthritis due to an immunological reaction. In such cases, pathogenicity due to circulating immune complex should be suspected. Anti-inflammatory treatments such as corticosteroids or immunosuppressive therapies are ineffective in cases of schistosomal arthropathy. A joint fluid puncture appears to be necessary and parasitic examination as well as in situ immunological techniques appear to be important in order to confirm the diagnosis of schistosomal arthropathy. Conclusions. The frequency of articular schistosomiasis is probably underestimated and should be sought when patients have unexplained polyarthropathy, as it can be an alternative diagnosis when patients have concomitant parasitic infections. These situations are common, whereas the association between unexplained inflammatory arthritis and a concomitant parasitic infection is rarely made. Unspecific rheumatism can lead to probabilistic treatments with many side effects, and looking for a parasitic aetiology could lead to repeated antiparasitic treatments and may avoid other immunosuppressive or corticosteroid therapies. With increasing travel and global migration, physicians need to be more aware of nonspecific symptoms that may reveal an atypical presentation of a tropical disease that can be treated easily, thus avoiding inappropriate immunosuppressive treatments.

## 1. Introduction

Schistosomiasis is one of the most prevalent parasitic diseases in the world, mainly prevalent in Africa. The World Health Organization estimates that 200 million people in the world have schistosomiasis. Most infections are subclinical, and although they may be unidentified at the acute stage, they may progress to chronic symptoms depending on whether eggs are trapped in deep tissues. It is important to diagnose schistosomiasis at the earliest stage in order to treat the parasite before the eggs are trapped, and to avoid symptoms due to reactions to eggs being released.

The symptoms of arthropathy can be unspecific, which makes the aetiology hard to identify precisely. In many situations, probabilistic treatments such as immunosuppressive therapy are initiated. Microorganisms such as bacteria and parasites can all cause acute or chronic arthritis, which can be divided into infective/septic or aseptic arthritis (reactive or inflammatory). Sometimes, no diagnosis is reached in the first aetiological search. However, rethinking the parasitic aetiology in cases of arthritis, based on the direct examination of joint liquid or serology, may help to develop better targeted treatment strategies. 

## 2. Material and Methods

We performed a systematic review of the medical literature relating to schistosomal arthropathy, with a focus on the biological techniques and methods used to assign the joint disease at the schistosomiasis infection. A literature search was performed using the PubMed database, in accordance with the Preferred Reporting Items for Systematic Reviews and Meta-Analyses (PRISMA) guidelines (Figure 1). We searched for all articles published prior to July 2021, using the combination of the MeSH terms “joint”, “arthropathy”, “rheumatism”, “*Schistosoma*”, “parasitic”, and “arthritis”. All articles relating to joint impairment and schistosomiasis were included. Cases and reviews of these cases were included, and genus identification for schistosomiasis was not a criterion for exclusion. Articles written in English, French or Spanish were included. As schistosomal arthropathy is rarely described, non-full-text articles are also included in order to list every case published on the topic. 

To illustrate this review, we present a case report highlighting recurrent arthritis evolving concomitantly with urinary schistosomiasis. Unfortunately, the patient was lost to follow-up, so we do not have enough evidence of the direct link between his arthritis and schistosomiasis. Efficiency of antiparasitic treatment on his arthritis would help to conclude and propose therapeutic options.

## 3. Review of the Literature on Articular Schistosomiasis

In the medical literature, several case reports of arthropathy concomitant with schistosomal infection have been reported, mainly in Egypt, a country in which *Schistosoma* is endemic. We distinguished between case reports with parasitic eggs found in situ (less common situations) and case reports with clinical patterns of arthritis or arthropathy related with a concomitant infection with *Schistosoma* spp. All case reports published prior to 2021 are summarised in Table 1. Of all these case reports, which included 241 patients with arthropathy associated with schistosomiasis, only 4 reported microbiological documentation in situ.

The main review on this topic was reported by Bassiouni et al. In it, the authors described 124 cases of joint impairment associated with *Schistosoma* [3] in an Egyptian study population. The inclusion criteria were a clinical pattern with arthropathy and a positive *Schistosoma* serology. Joint pain was mainly lumbar, without any loss of joint function, effusion, deformity or morning stiffness. Interestingly, *Schistosoma* infection was chronic (mean duration: nine years). The authors suggested that chronic infection with the recurrent release of eggs in deep tissue is frequently linked to inflammatory flare ups. The authors carried out synovial biopsies on the most painful joint when the patients had a high egg concentration in the urine. Of the 11 patients, on whom these procedures were performed, 3 presented *Schistosoma* spp. eggs in their biopsy, with 1 being typical of *S. haematobium*.

Atkin et al. describe 72 patients with joint pain, out of 96 patients, who were followed up for an *S. mansoni* infection diagnosed on rectal biopsy finding *S. mansoni* eggs [4] Clinical patterns were enthesitis and symmetrical peripherical arthritis. Evolution was favourable after treatment with PZQ for all of them. In total, 24 patients did not have any joint pain. A significant difference was noticed in the duration of schistosomiasis disease (mean duration of follow-up was 3.7 years for patients without articular pain, and 9.8 years for patients with articular pain). This cohort underlines a high prevalence of joint disease in patients with *S. mansoni* infection, especially when parasite disease remains for several years. The association between inflammatory diseases of the musculoskeletal system and infection with schistosomiasis may be underestimated even if other studies are needed to assert this prevalence. 

A case report revealed a parasitological documentation in situ in a 24-year-old man living in Egypt [6]. The clinical pattern was acute septic arthritis initially due to *Staphylococcus aureus*, but several relapses with pain and fever despite the administration of appropriate antibiotherapy. A synovial biopsy was performed, which revealed intra-articular *Schistosoma* eggs. X-rays of the neck of the femur found osteolytic lesions with sclerotic bone. After PZQ treatment, the patient regained a good range of hip movement for several months. 

More recently, Rakotomala et al. reported two cases of chronic joint pain in Madagascar [9]. They were two similar clinical patterns of chronic arthritis, mainly lumbar and sacroiliac, which were resistant to anti-inflammatory treatments. In both cases, a joint fluid puncture revealed inflammation but no parasitic eggs in the joint fluid. The serology for *Schistosoma* was positive in the joint fluid in both cases (precision of serology tests between ELISA or Immunoblot is not mentioned). The hypothesis of the passive transmission of antibodies from the serum to the joint fluid was not investigated. Pain disappeared after PZQ treatment, with no recurrence within one year. This clinical evolution led the authors to conclude that these were cases of parasitic arthropathy. X-rays were normal in both cases [9]. 

In all the other case reports, the authors report arthropathy concomitant to *Schistosoma* infection, with no effect of anti-inflammatory treatments, and positive evolution with PZQ treatments. Diagnoses were mainly assumed with a positive serology in the blood and clinical evolution under parasitic treatment, even when parasitic eggs were not found in the joint fluid. Indeed, the immunological response to *Schistosoma* infection in acute or chronic infections is incomplete, and there is no total elimination of the mature worm and its eggs. This physiological cycle can explain the immunoallergic pattern, evolving with recurrent arthritis over the course of several years [11].

In contrast, biological investigations were conducted to explain the pathogenicity of the joints. Circulating immune complex (CIC) was explored as the cause of the pathogenicity on the joints in several reports of *Schistosoma* infections [12,13,14]. Bebars et al. compared immune complex circulation during schistosomal infection, with ELISA serology in the serum of 100 patients in four different groups: 40 with schistosomiasis and articular lesions, 20 with schistosomiasis without articular lesions, 20 with undocumented inflammatory arthritis, and 20 negative controls [15]. They found a significative increase in schistosomal circulating immune complex in the blood when schistosomiasis was associated with articular lesions. Furthermore, circulating immune complex in the joint fluid was very high, and higher than in the blood, which points towards the pathogenic role of this intra-articular complex on arthritis and indicates the use of serological investigations on the joints to reach a diagnosis of arthropathy when patients are exposed to parasitic endemicity. 

## 4. Illustrative Case Report

In May 2019, an 18-year-old male with no particular medical history, originating from Guinea and resident in France for one year, was admitted to the Internal Medicine Department at the university hospital in Marseille, France, with widespread joint pain and swelling of the left knee and both wrists, which had been evolving for 10 days. Clinical examination revealed inflammatory polyarthritis involving left knee arthritis with a limited range of motion, and diffuse hand arthritis in the metacarpophalangeal and proximal interphalangeal joints. The patient had no fever, no sensory or motor dysfunction, no cutaneous wounds, no adenopathy or splenomegaly and no haematuria. Laboratory tests revealed inflammation with leucocytosis (8 × 10^9^/L; cut-off: 4–10 × 10^9^/L) without eosinophilia (0.43 × 10^9^/L; cut-off: 0.1–0.7 × 10^9^/L) and elevation of the C-reactive protein (190 mg/L; cut-off < 5 mg/L); liver enzymes and electrolytes were normal. 

Further explorations with computerised tomography scans (CT scan) of the chest, abdomen and pelvis revealed a hepatomegaly, bladder wall calcifications without malignancy and multistage osteosclerosis. A bladder ultrasound revealed a thickening of the bladder wall with calcifications. A bone scan (scintigraphy) revealed bilateral fixations on the hands, the trapeziometacarpal joints, shoulders, left knee and ankle. There were no fixations on the sacroiliac or spinal joints.

Aetiological investigation of this left knee arthritis included a positive HLA B27 genetic marker with no other autoimmune markers other than positive antinuclear antibodies for 1/640 (anti-dsDNA antibodies, rheumatoid factor, anti-CCP antibodies were all negative). A left knee joint fluid puncture revealed neutrophilia without eosinophilia and elevated total proteins (57.9 g/L). Bacterial cultures remained sterile. No schistosomal eggs were found in the joint fluid at direct examination. A real-time polymerase chain reaction (RT-PCR) test, targeting the *dra1* gene for *S. haematobium*, was negative in joint fluid [15,16]. On the other side, investigation of the bladder calcifications revealed a positive serology for schistosomiasis (ELISA, Novagnost, NovaTec) (total IgG:optic density 1.45; cut-off 0.3), confirmed by a positive *Schistosoma* immunoblot assay LDBIO kit (LDBIO Diagnostic, Lyon, France). *S. haematobium* eggs were found in urine samples confirmed by RT-PCR targeting the *dra1* gene. 

Empirical treatment with corticosteroids (30 mg/day) was initiated for suspected inflammatory disease after administration of deworming doses of ivermectin. One month later, joint pain remained, and a new joint puncture was performed which revealed an aseptic elevation of proteins (60 g/L) (Figure 2). Thanks to the diagnosis of urinary schistosomiasis, a specific treatment was initiated by oral praziquantel (40 mg/kg/day in a single dose). Evolution was initially favourable after this treatment with disappearance of joint pain. 

Unfortunately, swelling and pain in the left knee began again two months after treatment (patient still had corticosteroid therapy but decreased to 12.5 mg/day). Recurrence of arthritis in this patient was hard to attribute to either the corticosteroid decrease or schistosomal arthropathy insufficiently treated with only one single dose of praziquantel.

Taking into account this recurrent and corticosteroid-resistant arthritis with no definitive diagnosis, we performed experimental serological profiles comparing sera and paired samples of joint fluid using immunoblotting (IB) (LDBIO Diagnostic, Lyon, France). IB revealed a positive pattern with specific bands of *Schistosoma* immunological response (Figure 2).

In order to clarify this result, we tried to differentiate between a specific intra-articular reaction and a passive transfer of antibodies from the serum to the joint fluid using the Goldmann–Witmer coefficient. The Goldmann–Witmer coefficient is used to define the ratio between specific antibodies (for a pathogen) in two different environments. It is usually used to diagnose parasitic uveitis with antibody assays in the aqueous humour (AH), especially for ocular toxoplasmosis [17,18,19]. The Goldmann–Witmer coefficient is the ratio [specific IgG in AH/specific IgG in serum]/[total IgG in AH/total IgG in serum]. It is interpreted as follows: 0.5 to 2, no secretion of antibodies in situ; 2 to 4, possible secretion of antibodies in situ; and > 4, definitive diagnosis of the secretion of specific antibodies in the aqueous humour. In our case, the Goldmann–Witmer coefficient was 1.34 (Table 2). Consequently, we cannot conclude as to the in situ excretion of *Schistosoma* antibodies. This result points towards the passive transmission of antibodies through the articular membrane, and to reactive immunological arthritis rather than schistosomal arthritis in situ.

These biological findings could be an argument to treat again with praziquantel and avoid immunosuppressive treatments. Unfortunately, the patient was lost to follow-up during the COVID pandemic. Evolution of arthritis with repeated cures of praziquantel cannot be evaluated, and there are insufficient data to propose these immunological analyses for definitive diagnosis of articular schistosomiasis. 

## 5. Discussion

Arthropathies during, or consequently to, parasitic infections are rare but may be underestimated [11]. They can result from the presence of the parasite itself inside the joint, through contiguous impairment from another nearby localisation, or through reactional impairment with an immunoallergic mechanism reacting to a parasitic infection elsewhere [20].

In reactive arthritis cases, it is difficult to demonstrate the parasitic aetiology [20]. Clinical patterns are polymorphic, mainly mono- or oligoarthritis of major joints (such as the knee), that can indicate the beginning of a spondyloarthropathy. This can also affect small and peripheral joints that can initially resemble rheumatoid polyarthritis [21]. Doury et al. proposed several clinical and biological indications to diagnose parasitic rheumatism (Figure 3) [20]. 

In our case report, criteria 1, 2, 3, 5, 6, 7 and 8 were present. We can also associate our patient with parasitic rheumatism based on the Doury criteria. This report underlines the link between schistosomal infections and recurrent arthritis as reported in different series.

The Goldmann–Witmer coefficient pointed towards the passive transmission of antibodies through the joint membrane, but we need more data to propose this coefficient for diagnosis of parasitic arthritis. Antiparasitic treatment remains the best therapeutic option in the case of recurrent arthritis with the resolution of symptoms after treatment in many cases published on this topic. Unfortunately, our patient was lost to follow-up during his corticosteroid treatment, and we need more reports to set guidelines for treatment of parasitic arthritis.

This review underlines the importance of differentiating between arthritis due to an immunological reaction following an infection and a parasitic pathogenicity in situ causing joint impairments to look for appropriate treatment. Therapeutic approaches can change depending on the mechanism and clinical evolution. Repeated antiparasitic treatments could be an alternative to corticosteroid therapy, if inefficient, and suspicion of parasitic aetiology. 

The medical literature on joint impairments, in cases of *Schistosoma* infections, is dated (reports are mainly dated before 1990) at a time when microbiological diagnostic methods were less developed than today. This review aims to update this association between parasitic infection and articular involvement and underline the new microbiological tools we can use to specify the diagnosis. Travellers are exposed to schistosomiasis during swimming in contaminated areas. Schistosomiasis is endemic in many countries and, if it is assumed that travel and migration will continue to increase, in the wake of the COVID-19 pandemic, it can be assumed that the number of patients suffering from schistosomiasis will increase in European countries.

Arthritis is rarely described as a clinical manifestation of the disease and is almost never mentioned in reviews on schistosomiasis. Joint pain and polyarthralgia are unspecific and can be related to schistosomiasis in endemic areas. This association is probably underestimated. Our review underlines the fact that joint pain and polyarthralgia are unspecific but more associated with schistosomiasis than previously believed in certain endemic areas. 

It is hard to differentiate between in situ parasitic pathogenicity in the joints and inflammatory and reactional lesions following infection, which are the most common patterns. According to the articles on the subject, the trend is towards using joint fluid punctures to confirm the diagnosis [22]. However, even when parasitic eggs are not found in situ, immunoallergic arthropathy exists in schistosomal infections and immune circulating complex seems to have a direct pathogenicity on the joints. The organism’s immune response to eggs trapped in tissues can stimulate inflammation and a local cell-mediated delayed-type hypersensitivity response with tissue impairment. All of these can be responsible for the deterioration of the joints. The first-line examination remains serology and stool or urine examinations, but joint fluid punctures can be used to look for schistosomiasis. The direct examination of urine or stool samples is a sensitive assay when concerning people returning from countries with a high prevalence of schistosomiasis. The detection of antibodies is useful in situations where prevalence is lower or when parasitic examination is negative. Searching for an immune reaction in the joint fluid could guide clinicians and help them to use antiparasitic treatments to lead to a positive evolution on arthritis. In the reactive mechanism, the adaptation of the Goldmann–Witmer coefficient remains an alternative way of confirming this hard-to-diagnose aetiology. Comparing an immunoblot with another pathogen could be another technique for differentiating between the in situ secretion of antibodies and passive transmission if immunisation to another pathogen is present in the blood.

Treatment is indicated at all stages of schistosomiasis with a single dose of praziquantel but retreating with the same dose may be an option for people who continue shedding viable eggs than can provoke inflammatory reactions. Articular schistosomiasis should therefore be an alternative potential diagnosis when there is no aetiology of a clinical pattern with polyarthralgia or arthritis in travellers and patients arriving from an endemic area.

## Figures and Tables

**Figure 1 pathogens-11-01369-f001:**
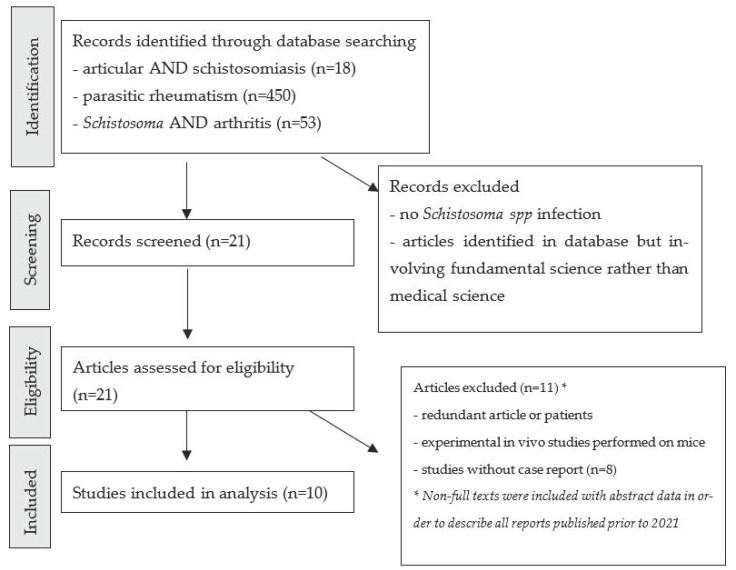
PRISMA diagram for review of the literature on articular schistosomiasis.

**Figure 2 pathogens-11-01369-f002:**
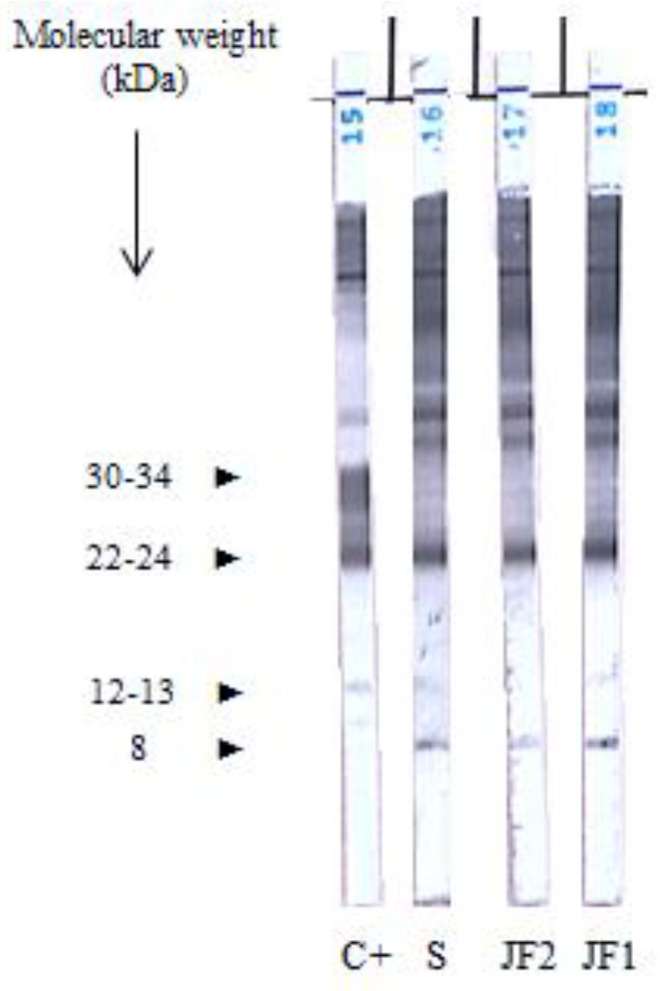
Immunological profile by IB for *Schistosoma* in serum (July 2019) (S) and joint fluid in May 2019 (JF 1) and July 2019 (JF 2), C + is the positive control.

**Figure 3 pathogens-11-01369-f003:**
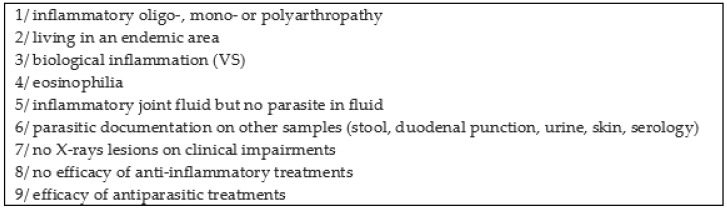
Doury criteria for articular impairment due to parasitic infection. According to Doury et al. (original table is in French), diagnosis is based on a body of clinical arguments. Criteria 1, 2, 6, 7, 8 and 9 are necessarily present for diagnosis. One of the best criteria for parasitic rheumatism is a good response to antiparasitic treatments.

**Table 1 pathogens-11-01369-t001:** Case reports on schistosomiasis and joint disease, review of literature between 1973 and 2021.

Publication (Reference)	Patients (Number)	Sex	Age (Years)	Country	Clinical Pattern	Positive Arguments for Schistosomiasis Arthritis	Urine or stool Sampling	Serology *Schistosoma*	Imaging	Treatment
Girges et al., 1966 [1]	36	NR	NR	Egypt	NR	NR		NR	NR	NR
May et al., 1973 [2]	2	M	NR	NR	NR	resolution after PZQ		NR	NR	PZQ 40 mg/kg
M Bassiouni et al., 1984 [3]	124	NR	10–30	Egypt	joint pain - 44% lumbar - 20% legs - 32% knees - 12% other joints - 10% heels	11 synovial biopsies performed 3/11 synovial biopsy revealed *Schistosoma* spp. eggs * (one biopsy = *S. haematobium*)		positive concomitantly	X-rays: sacroiliac lesions n 41% of cases:asymmetric irregular erosions in joint space, irregular sclerosisrare bone ankylosis	NR
L Atkin et al., 1986 [4]	72	40 M 32 F	21–40	Egypt	- 9: enthesitis, - 16: symmetric peripheral polyarthritis (PIP, MCP, wrists, knee, ankles, MTO) - 47: arthritis and enthesitis warm joints, not red. soft tissue swelling. morning stiffness	synovial biopsy in 4 patients: leucocytosis, no parasite eggs resolution after PZQ	positive stool: *S. mansoni eggs*	NR	NR	PZQ 40 mg/kg
Greenfield et al., 1986 [5]	1	M	35	Sierra Leone	inflammatory polyarthralgia with recurrent arthritis: knees, wrists, shoulders, temporomandibular, no fever	circulating immune complex in serum resolution after PZQ	stool or urine samples: no eggs	positive	X-rays of hands, feet and sacroiliac joints: normal	inefficiency of NSAIDs ** Resolution after PZQ *** three cures + thiabendazole
Fachartz et al., 1993 [6]	1	M	24	Egypt	septic arthritis due to *S. aureus* (treated with cephalosporine (cephradine) plus arthrotomy Persistent joint pain and fever	Synovial biopsy: eggs of *Schistosoma* spp. * resolution after PZQ	Urine and stool samples: no eggs	NR	X-rays: bone loss with void space CT: osteolytic lesions in femoral head and neck CT scan: calcification of bladder wall	Resolution three months after PZQ ***
Rolland et al., 1998 [7]	1	M	NR	France	NR	NR		NR	NR	NR
Lapa et al., 2011 [8]	1	F	26	Brazil	symmetric polyarthritis of wrists, MCP and PIP joints morning stiffness, asthenia for six weeks no fever, no diarrhoea	resolution after PZQ	positive stool: *S. mansoni* eggs	NR	X-rays: increased soft tissue in involved areas but no joint space narrowing or erosions	Resolution two weeks after PZQ *** Normal X-ray six months after treatment
Rakotomala et al., 2017 [9]	2	1 M1 F	32–42	Madagascar	chronic oligo-arthritis, mainly lumbar and sacroiliac painepisodic dysentery	positive serology in joint fluid in both cases, no microorganisms found in joint fluidresolution after PZQ		positive	X-rays: normal	Resistant to NSAIDs *** and sulfasalazine Resolution three weeks after PZQ *** three days
Saldarriaga RLM et al. [10]	1	F	45	Brazil	joint pain and swelling of the left shoulder, hands and knees synovitis of the wrists, PIP and MCP abdominal pain, diarrhoea and bleeding for two months	positive *S. mansoni* eggs on polyp biopsy (coloscopy) resolution after antihelminthic treatment	negative stool	NR	X-rays: normal echography: synovitis in wrists, MCP and PIP without erosions	Resistant to methorexate 7.5 mg per week and prednisone 5 mg per day Resolution after oxamniquina 15 mg/kg

M: male, F: Female, NR: not reported, MCP: metacarpophalangeal, PIP: proximal interphalangeal, MTO: metatarsophalangeal. NSAIDs: nonsteroidal anti-inflammatory drugs; CT scan: computed tomography scanner; X-rays: X-radiations; PZQ: praziquantel * parasite documentation in situ *** 40 mg/kg ** no precision about posology.

**Table 2 pathogens-11-01369-t002:** Goldmann–Witmer coefficient between joint fluid and serum from our patient.

	JF 1 (May 2019)	JF 2 (July 2019)	Serum
Serology *Schistosoma* spp. * OD	1.41	1	1.45
Ratio OD JF/serum	0.97	0.69	-
Total IgG (g/L) **	10.99	10.44	15.15
Ratio total IgG JF/serum	0.72	0.69	-
**GWC (ratio *Schistosoma* spp/ratio total IgG)**	**1.34**	**1**	-

OD: optic density. JF: joint fluid. GWC: Goldmann–Witmer coefficient * serology (ELISA Novalisa, Novagnost, NovaTec, Western blot LD-Bio France). Serology for these three samples was performed on the same day using the same assay. ** Total IgG was calculated by turbidity technique on Optilite The Binding Site^®^, The Goldmann–Witmer coefficient was calculated with the following formula: [OD specific IgG JF/OD specific IgG serum] over [total IgG JF/total IgG serum].

## Data Availability

Not applicable.

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
