# Peer review of "Schistosomiasis and Recurrent Arthritis: A Systematic Review of the Literature"

_pathogens, 2022, doi:10.3390/pathogens11111369_

Round 1
Reviewer 1 Report
Mortier et al bring an interesting study focusing on the direct envolvement of schistosomose infection in rheumatic pathology. This kind of studies are mportant and welcome for clinicians in general, as atypical clinical spectrum associated to schistosomiasis are not fully investigated nor disseminated, underestimating the true impact.
Overall, the MS is well written and clear for the readers. I recommend the MS publication after minor corrections.
-Fig. 2: provide an IB image with better resolution
- Please, uniform the references according to journal guidelines. Publications are cited differently, some are in full and others in the abbreviate form (i.e. Refs 2;7; 8…)
Author Response
Mortier et al bring an interesting study focusing on the direct envolvement of schistosomose infection in rheumatic pathology. This kind of studies are mportant and welcome for clinicians in general, as atypical clinical spectrum associated to schistosomiasis are not fully investigated nor disseminated, underestimating the true impact.
Overall, the MS is well written and clear for the readers. I recommend the MS publication after minor corrections.
Dear reviewer,
Thankyou for taking time to read and review my article.
-Fig. 2: provide an IB image with better resolution
Unfortunately, we don’t have other picture for the immunoblot since it is photography
- Please, uniform the references according to journal guidelines. Publications are cited differently, some are in full and others in the abbreviate form (i.e. Refs 2;7; 8…)
We uniformed the references to journal guidelines
Thankyou
Best regards,
Coline Mortier
Reviewer 2 Report
The manuscript addresses a little explored topic, joint involvement in the course of schistosomal infection, through a systematic review of the literature.
Some comments:
1. The text deserves careful review for errors, such as the one served in line 44 of the abstract: "...thus avoiding avoid inappropriate...."
2. As the manuscript deals with arthropathy supposedly associated with schistosomal infection, I suggest that throughout the text, preference is given to the term "schistosomal arthropathy" rather than "parasitic arthropathy".
3. I ask if there is a reference in the articles included in the systematic review to researching other causes of arthropathy before stating that the condition is necessarily associated with schistosomal infection.
4. In lines 122 to 125, how was the infection time evaluated?
5. When there was no parasitological diagnosis, that is, observation of Schistosoma eggs in feces or urine, how do the authors view the use of positive serology for the diagnosis? Remember that serological reactions can last for long periods of time positive after successful specific treatment.
6. In the authors' opinion, is the use of the Goldmann-Witmer coefficient justified in suspected cases of schistosomal arthropathy?
Author Response
Dear reviewer,
Thank you for reading my paper and taking time to review it. Here are my responses to your comments, and revisions on the manuscript are made in yellow.
I hope we answered to your questions.
Best regards
The manuscript addresses a little explored topic, joint involvement in the course of schistosomal infection, through a systematic review of the literature.
Some comments:
- The text deserves careful review for errors, such as the one served in line 44 of the abstract: "...thus avoiding avoid inappropriate...."
Modifications are done on the new manuscript.
- As the manuscript deals with arthropathy supposedly associated with schistosomal infection, I suggest that throughout the text, preference is given to the term "schistosomal arthropathy" rather than "parasitic arthropathy".
Modifications are done on the new manuscript when association with schistosomal infection is made. When discussing the “Doury criteria” that are used to any parasitic arthropathy, we prefer to leave this term as described by authors (P.Doury)
- I ask if there is a reference in the articles included in the systematic review to researching other causes of arthropathy before stating that the condition is necessarily associated with schistosomal infection.
Yes, we first reviewed all causes of parasitic arthropathy to compare if there was other manners to diagnose it or other reviews on non schistosomal parasitic arthropathy.
And then we focused on schistosomal infection to make a previse review of every described cases (and we illustrated the review with the case report we had)
- In lines 122 to 125, how was the infection time evaluated?
- When there was no parasitological diagnosis, that is, observation of Schistosoma eggs in feces or urine, how do the authors view the use of positive serology for the diagnosis? Remember that serological reactions can last for long periods of time positive after successful specific treatment.
Indeed, all articles we found report case-reports about association between joint pain or history of arthropathy concomitant to schistosomal infection, but no relation in time can be made in cases where there is no parasitological diagnosis. This difficulty to find a causal link is the point on which we wanted to insist to try to find other manners than serology for the diagnosis (use of PCR, immunological ratio, compared Western Blot in the joint fluid). In addition, literature on the subject is old and microbiological techniques were fewer than today.
- In the authors' opinion, is the use of the Goldmann-Witmer coefficient justified in suspected cases of schistosomal arthropathy?
We miss other experiences or reports on the subject to make a conclusion of using Goldmann-Witmer coefficient when suspected schistosomal arthropathy is made.
However, joint arthropathy can often lead to probabilistic treatment (immunosuppressive or corticosteroid) without proof of immunological disease. It is usually a bundle of arguments that direct the clinician to choose treatment.
Serology in joint fluid doesn’t need invasive supplementary action (it can be add on the joint puncture already made) and if the coefficient trends toward antibodies secretion in situ, it can be an argument to help the clinicians when diagnosis is not clear. We would propose to use it in these tricky situations.
Serology have limits in infectious diagnosis, and we definitively need more experiences and reports to use it in everyday life.
Thankyou for taking this time to review,
Best regards
Coline Mortier

Reviewer 3 Report
The association between schistosomiasis and arthritis is a subject little considered in the context of schistosomiasis due to the fact that many doctors are not aware of it. However, as mentioned in the articleby the authors, the numbers may be underestimated.Thus, I consider that the subject is of high relevance, especially in the context of tropical diseases, namely in schistosomiasis.
I consider that this document constitutes a good review of the literature, a compilation of all available information on this subject, wich is very important not only for medical doctors but also for the scientific and academic communities. I just found one more paper related to the subject.
The article includes an illustrative case report, an interesting case, showing the association between schistosomiasis and arthritis, reinforcing the need for a correct diagnosis and follow up in this clinical cases.
The methodology is well described and appropriate for this kind or article. The illustrative clinical case is well documented and discussed, resulting in well-founded conclusions. Figures and tables are a good complement for the article, making it easier to read.
The article is well written, well-structured and easy to understand.
Lines 193 and 210 - "Schistosoma" should be in italics or substituted by "schistosome"
Line 168 - "medecine" should be "medicine"
Author Response
Dear reviewer,
Thankyou for taking time to read and review my article.
We made modifications on « Schistosoma » and « medicine », they are in yellow in the manuscript
Best regards,
Coline Mortier
